# Symbolization, Prompt, and Classification: A Framework for Implicit Speaker Identification in Novels

**Yue Chen**[1][*] **Tian-Wei He**[2]**, Hong-Bin Zhou**[2]**, Jia-Chen Gu**[1]**, Heng Lu**[2]**, Zhen-Hua Ling**[1][†]

[1]National Engineering Research Center of Speech and Language Information Processing,
University of Science and Technology of China, Hefei, China
[2]Ximalaya Inc., Shanghai, China
cy321@mail.ustc.edu.cn, {gujc,zhling}@ustc.edu.cn,
{tianwei.he,hongbin.zhou}@ximalaya.com, bearlu007@gmail.com

## Abstract

Speaker identification in novel dialogues can be widely applied to various downstream tasks, such as producing multi-speaker audiobooks and converting novels into scripts. However, existing state-of-the-art methods are limited to handling explicit narrative patterns like "*Tom said, '...'*", unable to thoroughly understand long-range contexts and to deal with complex cases. To this end, we propose a framework named SPC, which identifies implicit speakers in novels via symbolization, prompt, and classification. First, SPC symbolizes the mentions of candidate speakers to construct a unified label set. Then, by inserting a prompt we reformulate speaker identification as a classification task to minimize the gap between the training objectives of speaker identification and the pre-training task. Two auxiliary tasks are also introduced in SPC to enhance long-range context understanding. Experimental results show that SPC outperforms previous methods by a large margin of 4.8% accuracy on the web novel collection, which reduces 47% of speaker identification errors, and also outperforms the emerging ChatGPT. In addition, SPC is more accurate in implicit speaker identification cases that require long-range context semantic understanding.

## 1   Introduction

Speaker identification in novel dialogues, also known as dialogue attribution (Muzny et al., 2017; Cuesta-Lázaro et al., 2022), aims at identifying the speaking characters of utterances in fiction texts (Glass and Bangay, 2007; Elson and McKeown, 2010). It is an important task for various downstream applications like automatically assigning appropriate voices to utterances in audiobook production (Pan et al., 2021) and novel-to-script conversion (Soo et al., 2019). As dialogues serve as

| Type | Speaker | Sentence |
|------|---------|----------|
| N1 | - | *Jin Xiu shouted with enthusiasm,* |
| U1 | Jin Xiu | *"Let's sing a song together!"* |
| U2 | Wu Zhongping | *"Old songs or new songs?"* |
| N2 | - | *Said Wu Zhongping.* |
| U3 | Lanxiang | *"You should say the songs of the present or the songs of the past."* |
| N3 | - | *Lanxiang corrected her friend with a laugh.* |
| U4 | Wu Zhongping | *"Well, you're right. For the songs of the past I only know Let's Sway Twin Oars."* |
| U5 | Gu Yangmin | *"It's just the time for it."* |
| N4 | - | *Said Gu Yangmin.* |

Table 1: A translated example from the Chinese novel *World of Plainness*. U1-U5 are utterances and N1-N4 are narratives.

the major interaction between characters in novels, automatic identification of speakers can also be useful for novel-based knowledge mining tasks such as social network extraction (Jia et al., 2020) and personality profiling of the characters (Sang et al., 2022).

Table 1 shows an example randomly sampled from the Chinese novel *World of Plainness*. For utterances U1, U2, U3, and U5, their speakers can be easily determined by recognizing the explicit narrative patterns like "*Said Wu Zhongping*" in the previous or following sentence. U4 is an exception that does not fall into this explicit pattern. To correctly predict the speaker of U4 requires an understanding of the conversation flow. Although many speaker identification cases can be solved by recognizing narrative patterns, many complex examples still call for a deep understanding of the surrounding context. We refer to these complex examples as implicit speaker identification cases. They pose difficulties for existing speaker identification algorithms.

---

[*]Work done during the internship at Ximalaya.
[†]Corresponding author.

Most recent approaches for speaker identification (Chen et al., 2021; Cuesta-Lázaro et al., 2022; Yu et al., 2022) are based on pre-trained language models (PLMs) like BERT (Devlin et al., 2019). PLM-based methods enjoy the advantage of the PLM's internal linguistic and commonsense knowledge obtained from pre-training. However, two main downsides should be highlighted for these methods. On the one hand, some methods truncate a context into shorter textual segments before feeding them to PLMs (Chen et al., 2021; Cuesta-Lázaro et al., 2022). Intuitively, this approach inevitably introduces a bias to focus on short and local texts, and to identify speakers by recognizing explicit narrative patterns usually in the local context of the utterance. It may fail in implicit speaker identification cases when such patterns are not available and long-range semantic understanding is indispensable. On the other hand, some methods adopt an end-to-end setting (Yu et al., 2022) that sometimes extracts uninformative speakers like personal pronouns. Besides, they only perform mention-level speaker identification in which two extracted aliases of the same character won't be taken as the same speaker. In recent months, large language models (LLMs) have become the most exciting progress in the NLP community. Although LLMs show impressive zero-shot/few-shot capabilities in many benchmarks (Patel et al., 2023; Ouyang et al., 2022; Chung et al., 2022), how well they work for speaker identification remains unknown.

Drawing inspiration from the successful application of prompt learning and pattern-exploiting training in various tasks like sentiment analysis (Patel et al., 2023; Schick and Schütze, 2021), we propose a framework to identify speakers in novels via **s**ymbolization, **p**rompt, and **c**lassification (SPC). SPC first symbolizes the mentions of candidate speakers to construct a unified label set for speaker classification. Then it inserts a prompt to introduce a placeholder for generating a feature for the speaker classifier. This approach minimizes the gap between the training objectives of speaker identification and the pre-training task of masked language modeling, and helps leverage the internal knowledge of PLMs. In previous studies, the inter-utterance correlation in conversations was shown to be useful for speaker identification in sequential utterances (He et al., 2013; Muzny et al., 2017; Chen et al., 2021). SPC also introduces auxil-

iary character classification tasks to incorporate supervision signals from speaker identification of neighborhood utterances. In this way, SPC can learn to capture the implicit speaker indication in a long-range context.

To measure the effectiveness of the proposed method and to test its speaker identification ability, SPC is evaluated on four benchmarks for speaker identification in novels, specifically, the web novel collection, *World of Plainness*, Jin-Yong novels, and CSI dataset. Compared to the previous studies that conduct experiments on merely 1-18 labeled books (Yu et al., 2022), the web novel collection dataset contains 230 labeled web novels. This dataset enables us to fully supervise the neural models and to analyze their performance at different training data scales. Experimental results show that the proposed method outperforms the best-performing baseline model by a large margin of 4.8% accuracy on the web novel collection. Besides, this paper presents a comparison with the most popular ChatGPT[1], and results indicate that SPC outperforms it in two benchmarks. To facilitate others to reproduce our results, we have released our source code [2].

To sum up, our contributions in this paper are three-fold: (1) We propose SPC, a novel framework for implicit speaker identification in novels via symbolization, prompt, and classification. (2) The proposed method outperforms existing methods on four benchmarks for speaker identification in novels, and shows superior cross-domain performance after being trained on sufficient labeled data. (3) We evaluate ChatGPT on two benchmarks and present a comparison with the proposed method.

## 2 Related Work

In recent years, deep neural networks have shown great superiority in all kinds of NLP tasks (Kim, 2014; Chen and Manning, 2014), including text-based speaker identification. Candidate Scoring Network (CSN) (Chen et al., 2021) is the first deep learning approach developed for speaker identification and outperforms the manual feature-based methods by a significant margin. For each candidate speaker of an utterance, CSN first encodes the shortest text fragment which covers both the utterance and a mention of the

---

[1]https://chat.openai.com/
[2]https://github.com/YueChenkkk/SPC-Novel-Speaker-Identification

candidate speaker with a BERT. Then the features for the speaker classifier are extracted from the output of BERT. In this way, the model learns to identify speakers by recognizing superficial narrative patterns instead of understanding the context. Cuesta-Lázaro et al. (2022) migrates a dialogue state tracking style architecture to speaker identification. It encodes each sentence separately with a Distill-BERT (Sanh et al., 2019) before modeling cross-sentence interaction with a Gated Recurrent Unit (Chung et al., 2014). Then a matching score is calculated between each character and each utterance. However, this method still just utilizes the PLM to model local texts and results in poor performance. Yu et al. (2022) adopts an end-to-end setting that directly locates the span of the speaker in the context. It feeds the concatenation of the utterance and its context to a RoBERTa (Liu et al., 2019; Cui et al., 2021), after which the start and end tokens are predicted on the output hidden states. Yet under end-to-end setting, only mention-level speaker identification is performed, in which two extracted aliases of the same character are taken as two different speakers.

Previous studies have shown the great advantage of applying deep-learning methods to speaker identification in novel dialogues, but these methods either are limited to recognizing superficial patterns or only identify speakers at mention level. Our study proposes a method to identify speakers based on context understanding and improves the performance on implicit speaker identification problems when long-range semantic information is needed. The proposed method outperforms other competitors given sufficient labeled data, and is more efficient in data utilization.

## 3 Methodology

### 3.1 Task Definition

Before we dive into the details of our proposed approaches, it would be necessary to declare the basic settings for the task of novel-based speaker identification. The sentences in the novel have been split and each sentence is identified as either an utterance or a narrative. The name and aliases of the speaking characters in the novel have been collected in advance. The occurrences of the speaking characters in the novel are referred to as **mentions**. For the **target utterance** whose speaker we intend to identify, a **selected context** that covers the target utterance is extracted and denoted as

$ctx = \{x_{-N_1}, ..., x_{-1}, u, x_1, ..., x_{N_2}\}$. $u$ denotes the target utterance and $x_{-N_1}, ..., x_{-1}, x_1, ..., x_{N_2}$ denote the $N_1 + N_2$ sentences surrounding $u$. The speaker of the target utterance is assumed to be mentioned in the selected context, while the exceptions (should be rare) are discarded from the dataset. [3] Assume that $m$ candidate speakers are located in $ctx$ by matching the text in $ctx$ to the speakers' names.

### 3.2 Framework Introduction

Figure 1 shows the framework of the proposed SPC. SPC takes the selected context as input and generates the likelihood of character classification as its output. First, the mentioned characters in the input context are replaced with symbols. Then, a prompt with a placeholder (the [MASK]) is inserted to the right of the target utterance. After that, placeholders of auxiliary tasks are introduced into the context. At last, the PLM encodes the processed context and classifies the characters at each placeholder.

### 3.3 Character Symbolization

Character symbolization unifies the candidate speaker sets in different contexts, after which speaker identification can be formulated as a classification task. We assign a unique symbol $C_j (j = 1, ..., m)$ to each candidate character mentioned in $ctx$, and replace the mentions of each character with its corresponding symbol in $ctx$. Note that this mapping from characters to symbols is only used for the specific selected context rather than for the whole novel, so as to reduce the number of classification labels. The character symbols form a local candidate set $CS = \{C_1, ..., C_m\}$. Let $M$ be the maximum number of candidate characters. $C_1, ..., C_M$ have been added to the special token vocabulary of the PLM in advance. The embeddings of these special tokens are randomly initialized and will be jointly optimized with other model parameters.

### 3.4 Prompt Insertion

Next, we insert into the selected context a prompt $p$="[prefix] ___ [postfix]", right after the target utterance. In the inserted prompt, "___" is the placeholder we aim to classify, while [prefix] and [postfix] are manually crafted strings on both sides of the placeholder. In practice, we choose

---

[3] In practice, we use a 21-sentence window. 97.2% of utterances in the web novel collection are included.

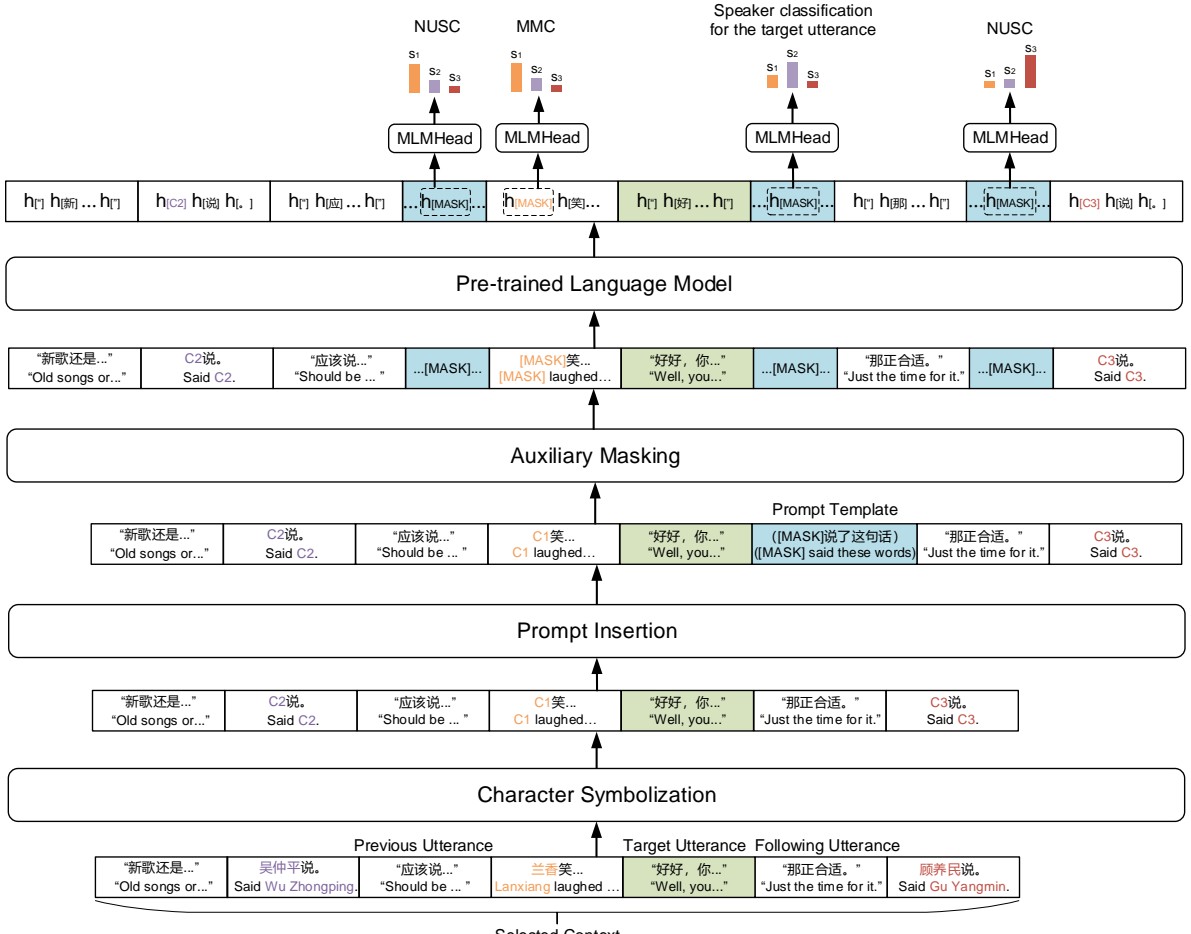

Figure 1: The framework of SPC. The block in green represents the target utterance, while the blocks in blue represent the inserted prompt templates. The uncolored blocks are other sentences in the selected context. The mentions of candidate speakers are also colored.

"（" and "说了这句话）" for [prefix] and [postfix] respectively, which combine to mean "(＿＿＿ said these words)" in English. The [MASK] token in the PLM's vocabulary is used for the placeholder.

### 3.5 Speaker Classification based on a PLM

Then we feed the processed context to a PLM which has been pre-trained on masked language modeling (MLM) to classify the missing character at the placeholder. Specifically, we utilize the pre-trained MLM head of the language model to produce a scalar score $s_j(j = 1, ..., m)$ for each candidate character $C_j$:

$$\boldsymbol{h}^{'} = \text{LayerNorm}(\text{GELU}(\boldsymbol{W}_T\boldsymbol{h} + \boldsymbol{b}_T)), \quad (1)$$

$$\boldsymbol{s} = \boldsymbol{W}_D\boldsymbol{h}^{'} + \boldsymbol{b}_D, \quad (2)$$

$\boldsymbol{h} \in \mathbb{R}^d$ is the output hidden state of the PLM corresponding to the placeholder, where $d$ is the hidden size of the PLM. $\boldsymbol{W}_T \in \mathbb{R}^{d \times d}$ and $\boldsymbol{b}_T \in \mathbb{R}^d$ are the weight and bias of the linear transform layer. $\boldsymbol{W}_D \in \mathbb{R}^{M \times d}$ and $\boldsymbol{b}_D \in \mathbb{R}^M$ are the weight and

bias of the decoder layer. $\boldsymbol{W}_T$ and $\boldsymbol{b}_T$ are pre-trained along with the PLM, while $\boldsymbol{W}_D$ and $\boldsymbol{b}_D$ are randomly initialized. $\boldsymbol{s} = [s_1, .., s_M]$ is the output score vector in which the first $m$ scores correspond to the $m$ candidate in $CS$.

At the training stage, as our goal is to assign the correct speaker a higher score than other candidates, we choose margin ranking loss to instruct the optimization. For example, assume the correct speaker of the target utterance is $C_j$, then $C_j$ is paired with each candidate in $CS \backslash C_j$. The speaker identification loss of $u$ is calculated on the scores of each candidate pair, as:

$$\mathcal{L}(u, ctx) \quad (3)$$

$$= \frac{1}{m-1} \sum_{k=1, k \neq j}^{m} max\{s_k - s_j + mgn, 0\},$$

where $mgn$ is the ideal margin between the scores of the two candidates. At the inference stage, the candidate assigned the highest score is identified as the speaker.

### 3.6 Auxiliary Character Classification

Based on the classification task form, we designed two auxiliary character classification tasks: 1) neighborhood utterances speaker classification (**NUSC**) to utilize the speaker alternation pattern between adjacent utterances, and 2) masked mention classification (**MMC**) to alleviate the excessive reliance on neighborhood explicit narrative patterns. NUSC teaches the model to classify the speakers of the neighborhood utterances of the target utterance. In this way, the model learns to utilize the dependency between the speakers of neighborhood utterances. MMC randomly masks the character mentions in the neighborhood sentences of the target utterance and quizzes the model on classifying the masked characters. It corrupts the explicit narrative patterns like "Tom said" which usually exists in the neighborhood sentence of the utterance and guides the model to utilize long-range semantic information.

We relabel the target utterance as $u_i$ and its previous and following utterance as $u_{i-1}$ and $u_{i+1}$. To perform NUSC, the prompt $p$ is also inserted after $u_{i-1}$ and $u_{i+1}$, as long as these two utterances and their speakers are covered in $ctx$. Then, the model classifies the speakers for $u_{i-1}$ and $u_{i+1}$ as described in Section 3.5. The loss introduced by NUSC is the average speaker classification loss of the two neighborhood utterances:

$$\mathcal{L}_{NUSC} = \frac{1}{2}\mathcal{L}(u_{i-1}, ctx) + \frac{1}{2}\mathcal{L}(u_{i+1}, ctx). \quad (4)$$

For MMC, we randomly mask the character mentions by a probability in the previous and following sentences of the target utterance (i.e. $x_{-1}$ and $x_1$), provided the masked characters are mentioned somewhere other than these two sentences. The model is required to predict the masked characters. Let the indexes of the masked characters be $MC$, then we have the average classification loss of each masked mention as:

$$\mathcal{L}_{MMC} = \frac{1}{|MC|} \sum_{j \in MC} \mathcal{L}_{mgn}(j), \quad (5)$$

in which $\mathcal{L}_{mgn}$ is the same margin ranking loss function as is shown in Eq.(3).

After incorporating NUSC and MMC into the training objective, the loss function turns into:

$$\mathcal{L}_{ACC} = \mathcal{L}(u_i, ctx) + \alpha\mathcal{L}_{NUSC} + \beta\mathcal{L}_{MMC}, \quad (6)$$

where $\alpha$ and $\beta$ control the strength of supervision signals from NUSC and MMC respectively. At the inference stage, we do not mask the character mentions in $x_1$ and $x_{-1}$ to retain the explicit narrative patterns. The prompt is inserted after $u_{i-1}$ and $u_{i+1}$ at the inference stage to keep training-inference consistency.

## 4 Experiments

### 4.1 Datasets

We conducted experiments on four Chinese speaker identification datasets which are the web novel collection (WN), *World of Plainness* (WP) (Chen et al., 2019) [4], Jin-Yong novels (JY) (Jia et al., 2020) [5], and end-to-end Chinese Speaker Identification dataset (CSI) (Yu et al., 2022) [6] respectively. WN is a large internal speaker identification dataset. Its annotation details can be referred to in Appendix D. WN and CSI both consist of web novels of various genres and writing styles, while WP and JY are printed literature with more normative writings. WP and JY can serve as cross-domain evaluation datasets for WN. As no test data was provided in CSI, we use the development set of CSI as the test data and randomly sampled 10% of instances from the training set for validation. The number of novels and utterances in each subset of each dataset is shown in Table 2. There are no overlapped novels in the different subsets of WN.

For clarity, each subset of a dataset is named as "dataset-subset", e.g., "WN-train" for the training set of WN. To evaluate the impact of a smaller training set, we also sampled 5 novels from WN-train to make a smaller training set with about 31k utterances. To distinguish between the whole training set and the sampled one, we referred to them as WN-large and WN-small respectively.

### 4.2 Baselines

We compared SPC to the following baselines.
**CSN** (Chen et al., 2021) feeds a candidate-specific textual segment to the PLM and then outputs the likelihood of the corresponding candidate. A revision algorithm based on speaker alternation pattern is adopted to revise the speaker identification results in two-party dialogues.
**DST_SI** (Cuesta-Lázaro et al., 2022) encodes each sentence separately with the PLM before

---

[4]https://github.com/YueChenkkk/Chinese-Dataset-Speaker-Identification

[5]https://github.com/huayi-dou/The-speaker-identification-corpus-of-Jin-Yong-novels

[6]https://github.com/yudiandoris/csi

| Dataset | Subset | # of books | # of instances |
|---------|--------|-----------|----------------|
|         | train  | 118       | 477,989        |
| WN      | val    | 42        | 214,854        |
|         | test   | 70        | 398,287        |
|         | train  |           | 2,000          |
| WP      | val    | 1         | 298            |
|         | test   |           | 298            |
|         | train  |           | 17,159         |
| JY      | val    | 3         | 5,719          |
|         | test   |           | 5,719          |
|         | train  |           | 43,233         |
| CSI     | val    | 3         | 4,804          |
|         | test   |           | 17,503         |

Table 2: Statistics of WN, WP, JY, and CSI datasets.

modeling cross-sentence interaction with a GRU. The embeddings of the character mention and the utterance are dotted to obtain the likelihood. A CRF is employed to model the dependency between speakers of neighborhood utterances. Due to limited GPU memory, we only use a base model for DST_SI.

**E2E_SI** (Yu et al., 2022) feeds the concatenation of the utterance and its context to the PLM, after which the start and end tokens are predicted on the output hidden states.

**GPT-3.5-turbo** was based on GPT-3 (Patel et al., 2023) and aligned to human preference by reinforcement learning with human feedback (Ouyang et al., 2022). We prompted the model with the format "{context}#{utterance}#The speaker of this utterance is:". We used a tolerant metric that the response was considered correct if the true speaker's name is a sub-string of the response.[7] We only evaluated this baseline on WP and JY due to the expensive API costs.

### 4.3 Implementation Details

As context selection plays a critical part in speaker identification, we detail the context selection procedures for the methods we implemented. For SPC, we selected a 21-sentence context window surrounding the target utterance, which corresponds to $N_1 = N_2 = 10$ in Section 3.1. If the context window exceeds the PLM's length limit (512 tokens for RoBERTa), we would truncate the context window to fit the input length requirement. Since DST_SI is not open source, we implemented it ourselves. We followed their paper and segment conversations by restricting the number of intervening narratives

---

[7]We also tried prompting with multiple choices of candidate names but the performance degraded drastically.

between utterances to 1. We further included the previous and following 10 sentences of each conversation, and limited the maximum number of involved sentences to 30. More details can be referred to Appendix A.

### 4.4 Overall Results

We tuned and evaluated the models on the same dataset (in-domain), or tuned the models on WN and evaluated them on WP and JY (cross-domain). Note that although we compared zero-shot GPT-3.5-turbo to other cross-domain results, it hadn't been tuned on any data. The released CSI dataset masked 10% of tokens due to copyright issues, so we collected E2E_SI's performance on the masked CSI from the GitHub page of CSI. Validation/Test accuracies are shown in Table 3. We will mainly discuss the test results and leave the validation results for reference.

First, we can conclude from the table that RoBERTa-large performed better than RoBERTa-base and BERT-base for the same method. Regardless of the specific PLM, the comparative relationship between different methods remains constant. So we mainly focus on the performance of different methods based on RoBERTa-large. SPC based on RoBERTa-large consistently performed better than or comparable to all non-LLM baselines in both in-domain evaluation and cross-domain evaluation. In the in-domain evaluation of WN, SPC outperformed the best opponent CSN by 4.8% and 3.9% trained on WN-large and WN-small, achieving overall accuracies of 94.6% and 90.0%. These are remarkable improvements as the errors are reduced by 47% and 28%. As WN-test includes 70 web novels of various genres, we believe it reflects general performance on web novels. In the in-domain evaluation on WP which is the only dataset evaluated in CSN paper (Chen et al., 2021), SPC still outperformed CSN by 1.0%. We observed MMC might cause serious overfitting for very small datasets like WP-train, so we didn't adopt MMC for WP-train. In cross-domain evaluations, SPC also consistently outperformed all non-LLM baselines, which shows its better generalizability to novels of unseen domains.

Although GPT-3.5-turbo underperformed WN-large tuned SPC, its zero-shot performance is still remarkable. In comparison, GPT-3.5-turbo has a much large number of parameters and benefits from its vast and general pre-training corpus, while

| | PLM | In-domain | | | | | Cross-domain | | | |
|---|---|---|---|---|---|---|---|---|---|---|
| Training data | | WN-large | WN-small | WP-train | JY-train | CSI-train | WN-large | WN-small | WN-large | WN-small |
| Test data | | WN-test | WN-test | WP-test | JY-test | CSI-test | WP-test | WP-test | JY-test | JY-test |
| E2E_SI (Yu et al., 2022) | RoBERTa-large | -/- | -/- | -/80.9† | 98.1†/98.3† | -/85.9† | -/- | -/- | -/- | -/- |
| CSN (Chen et al., 2021) | BERT-base | -/- | -/- | -/82.5§ | -/- | -/- | -/- | -/- | -/- | -/- |
| CSN | RoBERTa-base | 90.4/88.6 | 86.8/84.8 | 80.5/85.2 | 98.4/98.4 | 90.0/88.8 | 85.6/86.9 | 80.5/82.2 | 96.6/96.9 | 95.9/96.2 |
| CSN | RoBERTa-large | 91.8/89.8 | 88.6/86.1 | 85.2/88.9 | 98.8/98.7 | 91.1/**90.6** | 91.3/91.3 | 83.9/86.6 | 97.6/97.9 | 96.7/96.8 |
| DST_SI (Cuesta-Lázaro et al., 2022) | RoBERTa-base | 86.0/84.1 | 71.6/69.2 | 63.4/61.7 | 96.7/96.9 | 91.1/87.2 | 78.5/78.9 | 68.8/67.8 | 93.4/94.5 | 79.1/79.3 |
| SPC (ours) | RoBERTa-base | 94.3/93.2 | 88.7/87.2 | 83.2/86.2 | 98.5/98.6 | 90.3/90.0 | 89.9/91.6 | 80.9/83.9 | 98.5/98.5 | 97.0/96.9 |
| SPC (ours) | RoBERTa-large | 95.6/**94.6** | 91.6/**90.0** | 90.3/**89.9** | 98.8/**98.9** | 90.5/**90.6** | 93.0/**91.9** | 88.9/88.6 | 98.9/**98.8** | 98.3/**98.1** |
| GPT-3.5-turbo (zero-shot) | GPT-3.5-turbo | -/- | -/- | -/- | -/- | -/- | 90.3/89.6 | 90.3/**89.6** | 92.7/93.0 | 92.7/93.0 |

Table 3: Validation/Test accuracies (%) of SPC and the baselines (†: numbers reported in (Yu et al., 2022); §: number reported in (Chen et al., 2021)). The results of GPT-3.5-turbo were obtained under a zero-shot setting without using any training data. The cross-domain validation accuracies were obtained on WP-val/JY-val. The highest test accuracy in each column is emphasized in bold.

| Training data | WN-large | WN-small | WP-train | JY-train |
|---|---|---|---|---|
| Test data | WN-test | WN-test | WP-test | JY-test |
| SPC | 94.6 | 90.0 | 89.9 | 98.9 |
| - ACC | 93.8 | 89.4 | 86.2 | 98.7 |
| - Prompt | 93.2 | 84.6 | 74.3 | 98.7 |

Table 4: Evaluation accuracies (%) of SPC and its ablated methods. The indentation indicates each ablation is based on the previous ones. ACC stands for auxiliary character classification.

SPC excelled by effectively tuning on an adequate domain-specific corpus. It's worth mentioning that the response of GPT-3.5-turbo may contain more than one name, e.g., "Jin-bo's sister, Xiu" and "Run-ye's husband (Xiang-qian)". These responses may fool our evaluation criterion, as the response is only required to cover the true speaker.

## 4.5 Ablation study

We conducted ablation studies based on SPC to investigate the effect of each module, with results shown in Table 4.

We first removed ACC from SPC. As can be seen in the table, removing ACC dropped the evaluation accuracies by 0.8% and 0.6% on WN, by 0.2% on JY, and by 3.7% on WP. This indicates that the auxiliary character classification tasks are beneficial for improving speaker identification performance. [8] Only the NUSC task was applied to training on WP-train, and it contributed a lot to the performance on WP-test. We think it's because the writing of WP is more normative than the writing of novels in WN. The sequential utterances in WP

usually obey the speaker alternation pattern, which can be easily learned and utilized.

We further ablated prompting. To this end, we did not insert the prompt but extracted the CSN-style features from the output of PLM to produce the likelihood of each candidate speaker. After ablating the prompt-based architecture, the performance of models trained on WN-large decreased by 0.6%, whereas those on WN-small and WP-train decreased drastically by 4.8% and 11.9%. It shows that prompting is helpful for boosting performance in a low-resource setting and verifies our starting point for developing this approach. Prompting closes the gap between the training objectives of speaker identification and the pre-training task, which can help the PLM understand the task and leverage its internal knowledge. JY is the exception in which performance did not degrade after this ablation, although its training set only contains 15k samples. We believe this is because JY is too easy and lacks challenging cases to discriminate between different ablations.

To gain insight into how the performance of SPC and its ablated methods varies with the scale of training data, we trained them on varying numbers of novels sampled from WN-large and evaluated their performance on WN-test. To facilitate comparison, we performed a similar evaluation on CSN. As is shown in Figure 2, every method achieved better accuracy with more training utterances. SPC consistently outperformed the other methods on all evaluated training data scales. It's observed that ACC brought almost constant improvements as the training data grew. While the prompt-based architecture was more effective in low-

---

[8] We also conducted pilot experiments to use more neighborhood utterances for NUSC but gained no improvement. See details in Appendix B.2.

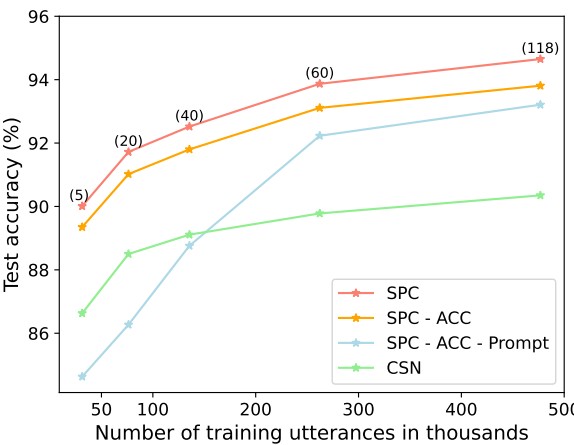

Figure 2: The speaker identification accuracies (%) of SPC, CSN, and the ablated methods of SPC on WN-test, trained on increasing utterances sampled from WN-large. The number in parentheses denotes the number of novels used for training.

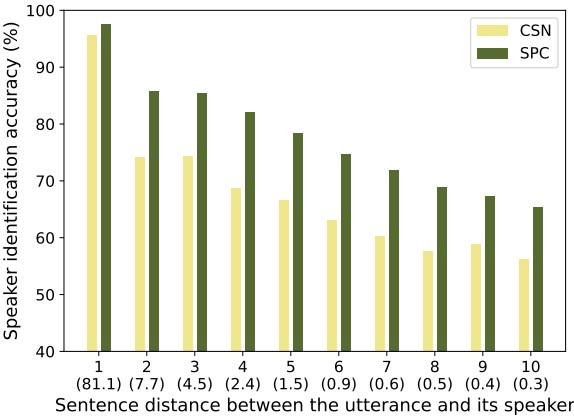

Figure 3: The speaker identification accuracies (%) of utterance groups divided by the sentence distance between the utterance and its speaker. The numbers in parentheses along the x-axis are the proportions (%) of utterances in the groups.

data scenarios and the improvement became less significant as training utterances increased.

The deeper ablated SPC (SPC w/o. ACC & prompt) initially under-performed CSN, but it overtook CSN when the training data reached about 150k utterances. In comparison, CSN didn't benefit much from the increment of training data. As a possible explanation, the deeper ablated method using a longer context window (445.5 tokens avg.) gradually learned to handle implicit speaker identification cases that require long-range context understanding, after being trained on more data. However, CSN using a short context window (131.3 tokens avg.) still failed in these cases. It's also revealed that generally more data is needed to train models that take a longer context window as input. However, with prompting and ACC, SPC overcomes this difficulty and learns to identify speakers in a long context window with a much smaller data requirement. As is shown in Figure 2, SPC took merely 31k utterances to reach the accuracy of 90%, while about 180k utterances were needed by its deeper ablated method to gain the same performance.

### 4.6 Implicit Speaker Identification Analysis

In this section, we're going to find out whether our proposed method shows a stronger ability to identify implicit speakers than other methods using a shorter context window, like CSN.

Intuitively, if the speaker's mention is close to the utterance, then the speaker of this utterance can probably be identified by recognizing explicit narra-tive patterns like "Tom said". On the contrary, utterances with distant speaker mentions are typically implicit speaker identification cases and require context understanding. We calculated the speaker identification accuracies of the utterances in WN-test, categorizing them based on the proximity of each utterance to the closest mention of its speaker. The comparison between SPC and CSN is shown in Figure 3. The term "sentence distance" refers to the absolute value of the utterance's sentence index minus the index of the sentence where the speaker's closest mention is found.

It can be seen from the figure that, as the sentence distance increased, both SPC and CSN identified speakers more inaccurately. Initially, at sentence distance = 1, both models performed comparably and achieved accuracies above 95%. However, when sentence distance came to 2, the identification accuracy of CSN drastically dropped to 74%. Utterances with sentence distance > 1 can be regarded as implicit speaker identification cases, so CSN is not good at identifying the speakers of these utterances. While SPC still maintained over 80% until sentence distance reached 5 and consistently outperformed CSN by 8.4% to 13.4% for sentence distance greater than 1. Thus it's verified that SPC has a better understanding of the context compared to CSN, and thus can better deal with implicit speakers. The proportion of test utterances at each sentence distance is also shown in Figure 3. 81.1% of the utterances are situated at sentence distance = 1, elucidating the reason behind CSN's commendable overall performance

despite its incapacity to handle implicit speakers.

We also conducted a case study on the implicit speaker identification cases in WN-test, with a few translated examples provided in Appendix C. SPC did perform better than CSN in many challenging speaker identification cases.

## 5 Conclusions

In this paper, we propose SPC, an effective framework for implicit speaker identification in novels via symbolization, prompt, and classification. Experimental results show SPC's superiority on four speaker identification benchmarks and its remarkable cross-domain capability. Furthermore, SPC significantly outperforms the strongest competitor CSN in implicit speaker identification cases that require deeper context understanding. We also evaluate ChatGPT on two speaker identification benchmarks and present a comparison with SPC. In the future, we hope to harness LLMs with longer input length limits to further improve speaker identification performance.

## Limitations

Although SPC has proved its effectiveness in novel-based speaker identification, we consider two aspects that can be further improved. First, we only implemented SPC on small base models containing less than 1 billion parameters. In light of the swift progress in LLMs, investigating the full potential of these advanced LLMs holds significant value and promise for future advancements. Second, in real-world applications, our approach operates on the output of a character name extraction module which might produce incorrect results. Thus, it's worth studying how to improve the robustness of the speaker identification approach and make it less vulnerable to errors in the character names.

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

## A Implementation Details

|  | SPC | CSN | DST_SI |
|---|---|---|---|
| Batch size | 128/8/32/128 | 192/12/32/32 | 64/4/16/16 |
| Learning rate | 1/1/4/4 e-5 | 1e-5 | 4e-5 |
| Max epoch | 50 | 50 | 50 |
| Early stop | 5 | 5 | 5 |

Table 5: Hyper-parameters for each method. Some parameters took different values for WN/WP/JY/CSI.

We adopted Chinese RoBERTa-wwm-ext-large and RoBERTa-wwm-ext-base [9] for CSN and SPC, while only the base model was adopted for DST_SI due to memory limitation. The large and base models contain 355M and 125M parameters respectively. We used BertTokenizer in Transformers toolkit [10] to tokenize the texts.

For SPC, we selected "（＿＿说了这句话）" as the prompt, which means "(＿＿ said these words)" in English. The maximum number of candidate speakers $M$ was set as 50. The training criterion of SPC has been demonstrated in Section 3 with the ideal margin $mgn$ set as 1.0. For SPC the auxiliary character classification loss factor $\alpha$ and $\beta$ in Eq.(6) were both set as 0.3. The mask probability for the masked mention classification task was 0.5. While the optimization criterion for DST_SI was minimizing the log-likelihood of the true speaker sequence. Adam optimizer (Kingma and Ba, 2015) was employed for optimization and the learning rate decayed every epoch by multiplying 0.98. The best models were selected by validation accuracy. Other hyper-parameters are shown in Table 5. Some of the parameters took different values for WN, WP, JY, and CSI respectively.

As the CSI dataset does not provide candidate speakers, we utilized the open-source NLP toolkit LTP (Che et al., 2021) [11] as a character name extractor to generate candidates.

We used 4 NVIDIA RTX 3090 GPUs for experiments. The GPU hours consumed to train each model for one epoch on WN-large is shown in Table 6. On WN-large the training of each investigated model converges in 30 to 50 epochs.

---

[9] The pre-trained model checkpoints can be obtained from https://huggingface.co/hfl/chinese-roberta-wwm-ext-large and https://huggingface.co/hfl/chinese-roberta-wwm-ext.

[10] https://github.com/huggingface/transformers

[11] https://github.com/HIT-SCIR/ltp

|  | SPC | CSN | DST_SI |
|---|---|---|---|
| RoBERTa-large | 16 | 39 | - |
| RoBERTa-base | 7.6 | 14 | 3.6 |

Table 6: GPU hours consumed in training one epoch on WN-large.

| [prefix] | [postfix] | Acc. (%) |
|---|---|---|
| （ | 说了这句话） | 91.6 |
| ( | said these words) | |
| （ | 说） | 91.6 |
| ( | said) | |
| （ | ） | 91.6 |
| （ | 是一个友善的人） | 91.5 |
| ( | is a friendly person) | |
| - | 说了这句话 | 91.4 |
| - | said these words | |
| - | - | 90.6 |

Table 7: Validation accuracies (%) on WN-val of different prompt templates. The English translations of meaningful prompts are provided. "-" represents an empty string.

## B Experiment Settings Discussion

### B.1 Using Different Prompts

We tried different prompt templates to see how they affect speaker identification performance. As can be seen from Table 7, different prompt templates don't affect the performance much, but using a empty prompt would hurt the performance. This indicates SPC's insensitivity to prompt selection.

### B.2 Using N-neighborhood Utterances

We conducted pilot experiments on using different numbers of neighborhood utterances for neighborhood utterance speaker classification (NUSC). We trained the models on WN-small, showing their validation performance in Figure 4. It's clear that using 1-neighborhood utterances for NUSC (our setting for SPC) brings some improvements, compared to 0-neighborhood (not applying NUSC). However, extending to more neighborhood utterances does not bring further improvements. A possible explanation is that the dependency between neighborhood utterances mostly lies in adjacent utterances, instead of distant ones.

## C Case Study

Figure 5 shows three translated examples from WN-test, with speaker identification results of different

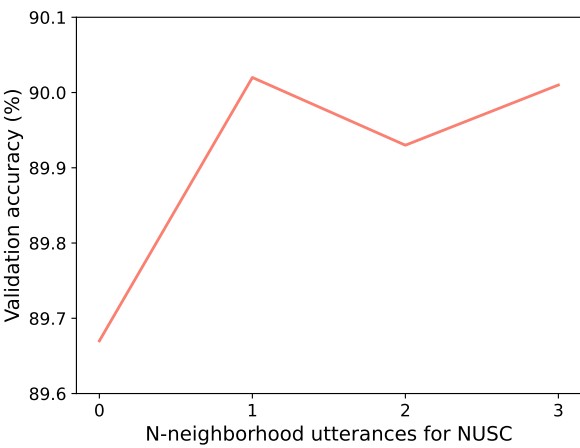

Figure 4: The validation accuracies (%) of using different numbers of neighborhood utterances for NUSC (w/o. MMC).

methods listed at the bottom of each example. They are all implicit speaker identification examples without explicit narrative patterns. We observed SPC did much better than CSN in solving these challenging speaker identification cases that require deeper context understanding. Example 3 shows that the auxiliary character classification tasks (ACC) help the model to better capture the inter-utterance dependency.

## D Web Novel Collection Details

The web novel collection was annotated on 230 web novels by a group of Chinese native annotators. The utterances in the novels were first identified based on quotes. Then, the annotators were instructed to mark the name of the speaker in the neighborhood context of each utterance, and the names of other characters were also marked if found.

Copyrights of the web novels belong to their respective proprietors. The authors are allowed to use the data for research purposes and should follow the principle of fair use. The data annotators are a group of employed professional data annotators each at least with a bachelor's degree. Their wages are in line with local regulations. Before the novels were handed to the annotators, they had been reviewed by a group of content reviewers to filter any offensive information, including violence, terror, abuse, etc. The annotation instruction informed the annotators of the potential use of the annotated data for speaker identification research.

## Example 1

1- Su Yue stood at the bottom of the steps, as Xiao Xiao looked at his black hair, something quietly revived in her heart, that had been deeply dormant in her heart from the first time she met him, she said to him softly: ",

2- "Su Yue."

3- He met her eyes and looked at her in silence, he was waiting for Xiao Xiao to speak, but she was silent for a long while, different from the past, as if thousands of thoughts came across her mind, until the steps were almost finished, she asked him gently:

4- "How does your family treat you?"

5- Su Yue suddenly stopped, he was shocked and raised his head, he never expected Xiao Xiao would suddenly ask him something like that.

**Who is the speaker of Sentence No. 4?**
**CSN:** Su Yue ✗
**SPC w/o. ACC:** Xiao Xiao ✓
**SPC:** Xiao Xiao ✓

---

## Example 2

1- "Melee, I just want to ask you now, what else did you lie to me about? Is it true that your mother is ill and in hospital?"

2- Melee opened her mouth slightly and shook her head slowly.

3- It seems that what she just said has all been heard by Lan An-xin.

4- "No, my mother is seriously ill, I'm not going to lie about it, I..."

5- "OK, Melee."

6- Lan An-xin lowered her eyes and did not want to see her again.

7- "An-xin, I'm sorry, I'm so sorry, I admit that some things I hid from you, I like Lu Shao-feng, for six years, I have no way to control ..."

8- "But you have already hurt me."

9- She spoke through gritted teeth.

10- For the first time, she feels betrayed by her friend.

11- "I'm sorry, really sorry, we can't be friends in the future, but we can also be partners. Please don't, we have to meet in the future."

12- "Meet?"

13- "Yeah."

14- Lan An-xin suddenly looked at her, she did not know what else to say.

15- At this point, can they meet again?

16- Finally, without saying anything, she picked up the notebook on the floor and left.

**Who is the speaker of Sentence No. 11?**
**CSN:** Lan An-xin ✗
**SPC w/o. ACC:** Melee ✓
**SPC:** Melee ✓

---

## Example 3

1- "Mo-xiang, come and do a simple hair style for me."

2- Pan-zhi whispered, her delicate face is a little pale, showing her weakness, but a bit of ruthless came across her eyes, Mo-xiang docilely stepped forward to help her comb the hair, and at this time, Hong-xiu pushed the door with tearful eyes, seeing Pan-zhi had already got up, she hastily asked with concern:

3- "Miss, you are not well yet, you shouldn't get up."

4- "Hong-xiu, the prince entered the palace, is it?"

5- Hong-xiu was startled, staring at Mo-xiang, this servant girl must be gossiping in Miss's ears, then she thought of the prince's purpose, her eyes got wet, suppressing the sorrow, she squeezed out a smile:

6- "Miss, the prince came to the palace only to meet the lord priminister."

7- "Hong-xiu, you don't have to hide anything from me, he has come here to break off the engagement, I definitely have to see him, how can he break it off without paying any price?"

8- There was a sense of chill in the words of Pan-zhi, Hong-xiu could not help but think of the scene where she forced the Servant Li to drink the poisoned wine, a shudder passed through.

**Who is the speaker of Sentence No. 4?**
**CSN:** Mo-xiang ✗
**SPC w/o. ACC:** Mo-xiang ✗
**SPC:** Pan-zhi ✓

Figure 5: Translated examples of implicit speaker identification cases from WN-test.