# OpenReview forum: "Symbolization, Prompt, and Classification: A Framework for Implicit Speaker Identification in Novels"
_EMNLP/2023/Conference — EMNLP 2023 Findings_

### Official Review · Reviewer_2pX6 · 2023-08-04

**Soundness:** 3

**Excitement:**

4: Strong: This paper deepens the understanding of some phenomenon or lowers the barriers to an existing research direction.

**Paper Topic And Main Contributions:**

The manuscript introduces a framework named SPC for identifying implicit speakers in novels via symbolization, prompt, and classification.
The key research problem addressed in this manuscript is implicit speaker identification in long range contexts such as narratives, for example. Existing state-of-the-art focus on handling explicit identification and hence do not do well in handling long-range contexts and complex cases. The proposed framework in this study uses symbolization, prompt and classification(SPC) to overcome and address this challenge. Experimental results presented shows that the method outperforms previous state-of-the-art by a large margin of 4.8% accuracy on the web novel collection and is more accurate in implicit speaker identification cases, where long-range context semantic understanding may also be required. The authors also showed that the proposed method outperforms ChatGPT in two benchmarks. The manuscript concludes with sufficient ablation studies showing the effects of each of the components of the framework as well as possible limitations of the approach.

**Reasons To Accept:**

1. The manuscript proposes an interesting framework named SPC for identifying implicit speakers in novels via symbolization, prompt, and classification.

2. Sufficient details on experimental setup was provided.

3. The manuscript was clear and concise.

**Reasons To Reject:**

Null

**Reproducibility:**

4: Could mostly reproduce the results, but there may be some variation because of sample variance or minor variations in their interpretation of the protocol or method.

**Reviewer Confidence:**

2: Willing to defend my evaluation, but it is fairly likely that I missed some details, didn't understand some central points, or can't be sure about the novelty of the work.

---

> ### Author Rebuttal · Authors · 2023-08-29
>
> We are truly grateful for your appreciation!

---

### Official Review · Reviewer_eb8A · 2023-08-04

**Soundness:** 3

**Excitement:**

3: Ambivalent: It has merits (e.g., it reports state-of-the-art results, the idea is nice), but there are key weaknesses (e.g., it describes incremental work), and it can significantly benefit from another round of revision. However, I won't object to accepting it if my co-reviewers champion it.

**Paper Topic And Main Contributions:**

The authors describe a framework for identifying speakers in novels by symbolizing the speakers entities into a set of symbols, generating a prompt template for masked pre-training, then predict the speakers via a classification task. Within the framework, the authors add two auxiliary tasks to improve the identification performance: neighborhood utterances speaker classification (NUSC) and masked mention classification (MMC). The experiments show that the proposed framework outperforms other baselines including the GPT-3.5-turbo LLM.

**Questions For The Authors:**

- It is not clear how the model can predict the speakers in the cross-domain dataset, since the framework first symbolize the speakers then predict the speaker by classification. Are the characters in the cross-domain dataset also in included in the symbol set?
- It is not clear how to run model inference on unknown data. If the character has aliases, how the model output will be normalized to a single speaker?

**Reasons To Accept:**

- The proposed method is able to identify the speakers of the utterances appearing in novel corpus.
- The model is able to generalize to cross-domain dataset with different writing styles.

**Reasons To Reject:**

- Whether the labels of NUSC and MMC tasks are from the original dataset or annotated by the authors is not clear. If they are annotated by the authors, the performance gain may be from the increase of the training data. It makes more sense to use those labels for other baselines for a fair comparison.

**Reproducibility:**

4: Could mostly reproduce the results, but there may be some variation because of sample variance or minor variations in their interpretation of the protocol or method.

**Reviewer Confidence:**

3: Pretty sure, but there's a chance I missed something. Although I have a good feel for this area in general, I did not carefully check the paper's details, e.g., the math, experimental design, or novelty.

**Typos Grammar Style And Presentation Improvements:**

page 2 col 2 line 128: neural models and to analyze their performance*s*
The bar charts in Figure 1 should be C1/C2/C3 instead of S1/S2/S3.

---

> ### Author Rebuttal · Authors · 2023-08-29
>
> 1. Re: ***The labels of auxiliary tasks***
>
> We didn't annotate additional data and the performance gain is not from the increase of annotated data. For NUSC, we used the previous and the following utterances of the target utterance for auxiliary speaker classification. NUSC is only performed if the speakers of the previous and the following utterances are already labeled in the original dataset. For MMC, we just masked the characters from the original text and used the model to predict them, which also doesn't require any annotation. We will clarify that in the revision. Thanks for your reminder.
>
> 2. Re: ***How can the model predict the speakers in unseen novels***
>
> For an excerpt from an unseen novel, as discussed in Section 3.1, we require the aliases of the characters in the novel to be collected in advance. Then, we only need to replace the characters in the excerpt with the symbols ($C_1$, $C_2$,...) and our model will work smoothly. The symbol $C_i$ represents the $i^{th}$ character appearing in the excerpt, rather than mapping to a specific character. We will enhance the clarity of Sections 3.1-3.2.
>
> 3. Re: ***How can the model output be normalized to a single speaker***
>
> Alias normalization occurs during character symbolization, through heuristic rules and manual correction. After the symbolization process, the different aliases are linked to the specific speaker and are replaced by the same symbol. We will clarify this process. Fully automated alias clustering is an interesting topic worth further exploration.
>
> 4. Re: ***Typos***
>
> Thank you for carefully reading our paper! We will revise the grammar error. As for Figure 1, the bars represent the prediction scores of the candidate speakers (see Section 3.5).

---

### Official Review · Reviewer_rtML · 2023-08-10

**Soundness:** 3

**Excitement:**

2: Mediocre: This paper makes marginal contributions (vs non-contemporaneous work), so I would rather not see it in the conference.

**Missing References:**

[The Project Dialogism Novel Corpus: A Dataset for Quotation Attribution in Literary Texts](https://aclanthology.org/2022.lrec-1.628/)

[Improving Automatic Quotation Attribution in Literary Novels](https://aclanthology.org/2023.acl-short.64.pdf)

[Measuring Information Propagation in Literary Social Networks](https://aclanthology.org/2020.emnlp-main.47)

**Paper Topic And Main Contributions:**

This paper proposes an approach to resolve speaker identification on novel dialogues, with a particular focus on the implicit case, where an utterance is not accompanied with explicit speaker cues, requiring strong understanding of the dialogue flow.. The approach first replaces speaker mentions of different context with special symbols to have a unified speaker label set, followed by RoBERTa encoding and speaker classification upon the hidden state. Two auxiliary tasks are also introduced in training to facilitate the model learning. Experiments are conducted on three datasets that compare against three baselines, and the proposed approach shows improvement most of the time. In addition, ChatGPT is also evaluated, which shows on-par performance with zero-shot prompting.

**Reasons To Accept:**

The task, experiments and analysis are described clearly; the proposed approach is able to surpass baselines most of the time, especially with improvement on implicit cases, which fulfills the motivation.

**Reasons To Reject:**

Although the proposed approach does appear better than baselines on the experimented datasets, there are three aspects that I found limiting its significance:

1. The proposed approach largely echos well-established insights from previous works, rather than presenting a new perspective. Including:
    1. The proposed auxiliary tasks (or some variants) are common in many dialogue-understanding tasks, such as dialogue-act prediction or dialogue language model pretraining.
    2. The prediction of speaker symbols does not have core difference from previous works such as E2E_SI, as they are all designed to predict the best speaker mention from the input, with varying model engineering or formality changes. Another similar example would be BookNLP that also predicts the best speaker mention.
    3. The add of prompting in RoBERTa is merely a simple engineering work.
2. The improvement of the proposed approach does not surpass baseline significantly, especially for the cross-domain setting, where the proposed approach has <1% improvement on 2 out of 4 evaluation sets. The most improvement is shown on WP-test; however, it then fails to offer better predictions than the zero-shot ChatGPT.
3. ChatGPT shows impressive zero-shot performance, which is essentially on a par with supervised models, and even achieves SOTA on WP-test directly. Although there appears a gap on JY, it is unclear that whether ChatGPT can be further improved through some simple tuning such as adding CoT. The paper does not provide details on how ChatGPT is prompted. Nevertheless, the proposed approach does not offer meaningful advantages under the era of LLMs.

In addition, there are other similar datasets in English focusing on speaker attribution. See Missing References.

**Reproducibility:**

4: Could mostly reproduce the results, but there may be some variation because of sample variance or minor variations in their interpretation of the protocol or method.

**Reviewer Confidence:**

5: Positive that my evaluation is correct. I read the paper very carefully and I am very familiar with related work.

---

> ### Author Rebuttal · Authors · 2023-08-29
>
> 1. Re: ***The auxiliary tasks are not novel***
>
> As demonstrated in Section 3.6, we introduce two auxiliary tasks: 1) neighborhood utterances speaker classification (NUSC) to utilize the speaker alternation pattern between adjacent utterances, and 2) masked mention classification (MMC) to alleviate the excessive reliance on neighborhood explicit narrative patterns. The two auxiliary tasks are introduced to tackle the specific challenges in training speaker identification models, and they enhance the performance. Unlike traditional dialogue tasks, novel-based speaker identification relies predominantly on understanding narrative context rather than solely on comprehending dialogues. We will address your concerns by conducting further investigations and comparing with existing works. Thanks for your reminder.
>
> 2. Re: ***SPC handles the similar task of E2E_SI and BookNLP***
>
> Indeed, E2E_SI and the open-source toolkit BookNLP can also serve for speaker identification. Our unique contribution lies in introducing a novel speaker identification framework--SPC--which surpasses the performance of the top baseline, particularly in implicit speaker identification scenarios.
>
> 3. Re: ***The add of prompting is a simple engineering***
>
> While we acknowledge that the addition of prompting alone may seem minor, it plays a pivotal role in our novel and robust framework for speaker identification. By incorporating the prompt, we not only introduce a placeholder for speaker symbol classification but also distinguish the target utterance from other utterances. We will highlight the significance of the prompt in our revision.
>
> 4. Re: ***The improvements are not significant***
>
> As shown in Table 3, our proposed SPC shows significant improvements in WN-test which is the largest test dataset. In cross-domain evaluation, you mentioned that SPC only absolutely surpasses CSN by 0.9% (indeed <1%) on JY-test. However, the accuracy of SPC and CSN is 98.8% compared to 97.9%, resulting in a 43% reduction in error rate. We will emphasize the improvements in terms of error rate.
>
> 5. Re: ***SPC is meaningless under the LLM era***
>
> At present, SPC exhibits greater robustness than LLMs and has the potential to be integrated with them to achieve even higher performance.
>
> We observed that the persistence of hallucination remains evident within LLMs. This is evident in the large performance gap observed between GPT-3.5-turbo and SPC on JY (see Table 3), a relatively easier benchmark compared to WP. We experimented with various prompting strategies in Section 4.2 and selected the best one. However, despite showing great potential in the speaker identification task, LLMs currently fall short of SPC's performance when provided with sufficient training data. Within our research budget, we will also continue exploring alternative prompts and reporting meaningful results.
>
> Furthermore, SPC is characterized by its data efficiency (as discussed in Section 4.5), suggesting its capability to reduce the resources needed (in terms of labeled data and computation) for effectively adapting LLMs to the speaker identification task.
>
> 6. Re: ***Missing evaluation on English datasets***
>
> Our paper focuses on the robustness and cross-domain adaptability of our proposed speaker identification framework in a single language, while multilingual adaptability is not one of our contributions. We also noticed there are English datasets for speaker identification, but these fall beyond the scope of this paper. We will consider adding these evaluations in the future.

---

### Official Review · Reviewer_M73c · 2023-08-20

**Soundness:** 3

**Excitement:**

3: Ambivalent: It has merits (e.g., it reports state-of-the-art results, the idea is nice), but there are key weaknesses (e.g., it describes incremental work), and it can significantly benefit from another round of revision. However, I won't object to accepting it if my co-reviewers champion it.

**Paper Topic And Main Contributions:**

The article presents a novel system for attributing written utterances to characters. The system relies on three stages: symbolization, prompt, classification. Its performance is argued to be superior to that of earlier systems, as measured on larger data than has been used in the past.

**Questions For The Authors:**

line 104: many readers may not understand the "minimizes the gap" argument.

In section 4.5, the authors draw some conclusions from differences observed on the WP test data (versus the WN test data). But according to Table 2, WP is tiny. There may be many novels in WN which are just like the single novel in WP. Why is this  special? Perhaps the authors should show a histogram over scores for individual novels in WN, and place the WP novel score in that context?

**Reasons To Accept:**

The proposed system appears to be evaluated on a larger variety of data than have been used for the same purpose in the past. The authors do a suitable job of comparing their system to the work of multiple previous authors.

**Reasons To Reject:**

The article is written insufficiently clearly, and understanding difficulties due to lack of clarity are likely to accumulate in the minds of readers, probably overwhelming many by the time the end of the article is reached. It seems that a large part of this problem is due to missing and crisp definitions of the terms "speaker", "mention", and "occurrence". (I think I understand that in a novel excerpt in which a dialogue between Fred and Nancy contains one reference to Fred as "Fred" and two references to him as "the boy", the number of speakers is 2, the number of mentions corresponding to Fred is 2, and the number of occurrences corresponding to Fred is 3. The text states on line 207 that mentions are occurrences, but on line 084 "mention" must be understood to mean "(unique) mention" rather than occurrence.) In addition, Table 2 and line 63 refer to "instances", which is not defined but could mean "occurrences". Finally, the authors use the phrase "characters are mentioned" (line 328) with "mention" as a verb, and it is not clear whether they mean "(unique) characters are mentioned" or "character mentions are mentioned". Section 3.3 is likely to be misunderstood (or just not understood) by most readers, and it is the first stage of the authors' pipeline. It is also not possible to understand the authors' definition of distance in the analysis part of the article.

The article also fails to provide compelling evidence of precisely which features of written dialogue/multilogue are handled better by their system than by existing systems. The ablation experiments -- although very welcome -- indicate which components of their proposed system are responsible for how much improvement, but not which features of written dialogue correspond to those components. I think the paper would be much stronger if the authors gave an example of dialogue/multilogue where "short-range semantic understanding" is necessary, where "long-range pragmatic (presumably this is what the authors mean when the say "patterns") understanding" is necessary, etc.

**Reproducibility:**

3: Could reproduce the results with some difficulty. The settings of parameters are underspecified or subjectively determined; the training/evaluation data are not widely available.

**Reviewer Confidence:**

2: Willing to defend my evaluation, but it is fairly likely that I missed some details, didn't understand some central points, or can't be sure about the novelty of the work.

**Typos Grammar Style And Presentation Improvements:**

The acronyms in Figure 1 should be resolved in its caption, since the figure is mentioned on line 220 on page 3; some (but not all) of those acronyms are resolved only o line 299 on page 5.

line 388: "x is dotted with y" could be replaced by the more commonly used phrase "to take the dot product between/of x and y".

line 406: "expensive API costs" -> "high API costs"

lie 431: "non-LLM baselines": which ones are those? the authors could make this more onbious to all readers

The authors use of the label "ChatGPT" is limited to the abstract, introduction, and conclusion; everywhere else the label "GPT-3.5-turbo" is used, presumably to mean the same thing. It may be safer to not assume equality between these two labels.

---

> ### Author Rebuttal · Authors · 2023-08-29
>
> 1. Re: ***The definition of "speaker", "mention", and "occurrence"***
>
> We genuinely appreciate your thorough review of our manuscript, and we sincerely apologize for any confusion our work might have caused. In our task definition section (Sec 3.1), we highlighted that the "occurrences" are referred to as "mentions". Thus in your Fred example, the number of mentions/occurrences is 3. In an excerpt, each potential speaker has one or more mentions/occurrences. The "unique mention" should be another word--"alias". The reason why we refer to "occurrence" as "mentions" is to be consistent with the terminology of other papers [1][2]. We apologize that "mention-level" in line 084 is indeed misleading and we will replace it with "alias-level". In addition, we will replace the term "mention" with "occurrence" in Section 3 for clarity. Specifically, the verb "mention" in line 328 can be replaced by "occur".
>
> 2. Re: ***The definition of "instance"***
>
> An "instance" in Table 2 and line 363 refers to an utterance whose speaker we aim to identify, not an "occurrence". We will add the definition of "instance" in the revision.
>
> 3. Re: ***The definition of "distance" in Section 4.6***
>
> In Section 4.6, specifically at line 552, we introduce the term "sentence distance". This metric is calculated as the absolute difference between the index of the utterance's sentence and the index of the sentence containing the nearest occurrence of the speaker. The concept of "sentence distance" serves as an indicator of the proximity between the speaker and the utterance, and it plays a pivotal role in quantifying the level of implicitness within speaker identification cases. This critical analysis experiment serves as validation for our key contribution. We are committed to improving the clarity of this section.
>
> 4. Re: ***Fail to show what features of written dialogues are better handled***
>
> In this paper, our main focus is to deal with the implicit speaker identification cases. We introduce the term "implicit speaker identification" in the paragraph starting at line 046, referring to utterances lacking explicit narrative patterns such as "... Said Tom". Section 4.6 presents a quantitative analysis of our proposed SPC's improvement in implicit speaker identification cases. We also show some implicit speaker identification examples in Appendix C, demonstrating how our proposed approaches can better solve these cases. We will include more examples to show more features of the implicit speaker identification cases. Thanks for the advice!
>
> 5. Re: ***Minimize the gap***
>
> In the context of line 104, we wish to illustrate that in our introduced framework, we formulate speaker identification as a masked language modeling (MLM) task which is exactly the pre-training task of some PLMs. We will enhance its clarity to ensure easier comprehension for other readers.
>
> 6. Re: ***WP vs. WN***
>
> In Section 4.1, we introduce the difference between WP and WN. WP is a published literature work, while WN is a bunch of online novels with less formal writing styles. Besides, as WP is a widely used data resource in previous papers [2][3], we have to report results on WP for comparison. We will show a histogram over scores for individual novels in our revision. Thanks for the advice!
>
> 7. Re: ***Typos***
>
> Thank you for your kind and valuable advice for improving our writing. We really appreciate your advice and will adopt them in our revision.
>
> References:
>
> [1] He H, Barbosa D, Kondrak G. Identification of speakers in novels. In Proceedings of the 51st Annual Meeting of the Association for Computational Linguistics (Volume 1: Long Papers) 2013 Aug (pp. 1312-1320).
>
> [2] Chen Y, Ling ZH, Liu QF. A Neural-Network-Based Approach to Identifying Speakers in Novels. InInterspeech 2021 (pp. 4114-4118).
>
> [3] Yu D, Zhou B, Yu D. End-to-End Chinese Speaker Identification. In Proceedings of the 2022 Conference of the North American Chapter of the Association for Computational Linguistics: Human Language Technologies 2022 Jul (pp. 2274-2285).

---

### Meta-Review · Area_Chair_p9FB · 2023-09-25

**Recommendation:** 3

**Metareview:**

This paper proposes a model for attributing dialogue utterances in a novel to the characters in that novel, in particular handling cases without explicit speaker cues.  Overall, the reviewers agree this is a decent contribution with appropriate experimental support, but with somewhat limited innovative quality.  The comparison with ChatGPT is interesting, especially in cases where it is outperformed by the proposed SPC model.

---

### Meta-Review · Senior_Area_Chairs · 2023-10-05

**Recommendation:** 3

**Metareview:**

meta

---

### Decision · Program_Chairs · 2023-10-07

**Decision:**

Accept-Findings

**Comment:**

This paper proposes a model for attributing dialogue utterances in a novel to the characters in that novel, in particular handling cases without explicit speaker cues.  Overall, the reviewers agree this is a decent contribution with appropriate experimental support, but with somewhat limited innovative quality.  The comparison with ChatGPT is interesting, especially in cases where it is outperformed by the proposed SPC model.|meta